# Antimicrobial stewardship interventions in least developed and low-income countries: a systematic review protocol

Grace Wezi Mzumara [1,2] Michael Mambiya,[1] Pui-Ying Iroh Tam [1,2,3]

[1]Child Health, Malawi-Liverpool-Wellcome Trust Clinical Research Programme, Chichiri, Blantyre 3 Malawi, Malawi
[2]Peadiatrics and Child Health, University of Malawi College of Medicine, Chichiri, Blantyre 3 Malawi, Malawi
[3]Liverpool School of Tropical Medicine, Liverpool, UK

**Correspondence to**
Dr Grace Wezi Mzumara;
gmzumara@mlw.mw

## ABSTRACT

**Introduction** Antimicrobial resistance (AMR) is increasing in low resource settings. It complicates the management of infectious diseases and is an increasing cause of death. This is due to, among other things, lack of health resources for appropriate diagnosis and unregulated access to antimicrobials in the public sphere. Developing context-specific interventions that enable judicious use of antimicrobials is important to curb this problem.

**Methods** We will conduct a systematic review of antimicrobial stewardship (AMS) approaches in Development Assistance Committee in least developed and low-income countries. The inclusion criteria are antimicrobial stewardship interventions in hospitalised patients of all age groups and exclusion criteria are community-based trials and studies that solely focus on viral, fungal or parasite infections. Antimicrobial stewardship interventions will be classified as structural, enabling, persuasive, restrictive or combined. Outcomes of included studies will be classified as clinical, microbiological or behavioural outcomes. The studies to be included will be randomised controlled trials, controlled before–after studies, interrupted time series trials, cohort and qualitative studies. Data will be extracted using forms adapted from the Cochrane collaboration data collection form. This systematic review will be conducted according to the Preferred Reporting Items for Systematic Reviews and Meta-Analyses guidelines and risk of bias will be done according to the Integrated quality Criteria for Review of Multiple Study Designs.

**Ethics and dissemination** Our findings will be presented to clinicians and policymakers, to support developing AMS protocols for low resource settings. We will publish our results in peer-reviewed journals.

**Trial registration number** CRD42020210634.

## Strengths and limitations of this study

► This study will identify antimicrobial stewardship (AMS) practices in low-income and least developed countries.
► We will describe the outcomes and impact of identified AMS practices.
► Risk of bias will be assessed using the Integrated quality Criteria for Review of Multiple Study Designs.
► Expected limitations include the variation in outcome measures, differences in the quality of the studies identified and publication bias.

were resistant to one or two first-line antibiotics.[7] Among children aged less than 5 years, resistance to first-line antibiotics from Gram-negative organisms rose from 3.4% to 30.2% and from 5.9% to 93.7% for Klebsiella species.[6] Drug-resistant Gram-negative bacteria are particularly concerning as they are associated with increasing mortality in low-income countries.[5 8 9]

AMR complicates the management of common infectious diseases in these settings. For example, the majority of cases and deaths from bacterial meningitis occur in children from less developed countries.[3] However AMR among antimicrobials used to treat *Streptococcus pneumoniae, Haemophilus influenza* or *Neisseria meningitidis* is found at presumed higher rates in low resource countries.[3 10] Other examples are diarrhoeal diseases that are the leading causes of death and disability in developing countries where enteric pathogens are found to be resistant to commonly prescribed antimicrobials.[1 11 12] Among neonates, the rising exposure to prophylactic antibiotics in pregnant women and very low birth weight infants raises concerns about possible AMR-resistant neonatal sepsis, which could increase cases and deaths from Group B Streptococcus.[2 8 13–15]

Several factors contribute to AMR in developing countries. First, the lack of financial resources and expertise for blood culture and

## INTRODUCTION

Infectious diseases are among the top leading causes of death worldwide and predominantly affect children in low and middle-income countries.[1–5] A major challenge to managing infectious diseases in these settings is the growing resistance to current antimicrobials.[6]

A large surveillance study in Malawi found that antimicrobial resistance (AMR) to first-line antibiotics (amoxycillin, chloramphenicol and cotrimoxazole) increased between 1998 and 2016.[7] About 40% of all isolates

antimicrobial sensitivity analyses is a challenge to appropriate prescription and surveillance.[8 10 11 13] Second, the use of prescription antibiotics is often unregulated and antibiotics can be obtained without prescriptions in many developing countries.[8 13 16] In these settings, cheaper alternatives are used in place of effective antibiotics and result in AMR in hospitals and communities.[13] This can cause changes in the pathogen profile of a population and perpetuate AMR.[8 13] Third is the lack of appropriate policies to regulate prescribing practices, which means that efforts to mitigate the problem are uncoordinated and ineffective.[8 16] Efforts to mitigate these challenges are important to prevent cases and deaths from infections attributable to AMR.

These efforts are antimicrobial stewardship (AMS) practices, which are coordinated interventions aimed at promoting appropriate prescription of antibiotics.[17 18] Structural interventions involve new diagnostic tests, persuasive interventions target behavioural change in prescribing practices, enabling interventions educate prescribers and restrictive interventions involve regulations for the usage of some antimicrobials.[17 18] AMS strategies can use these classes of interventions individually or in combination.[17 18] Other measures have incorporated clean water and hygiene practices for infection prevention.[8 13] All AMS strategies use guidelines with defined recommendations measured as quality indicators for each intervention.[16 18 19]

The WHO recommends surveillance systems for AMR to detect bacterial infections, identify resistance patterns and monitor interventions.[20] To do this, context-specific policies or protocols should be developed in order to guide acceptable and effective AMS solutions.[16–18 21] Although studies of different AMS practices have shown that they work to improve prescribing practices, there is little evidence to guide AMS in low resource settings.[4 17 18]

A review of AMS interventions in low and middle-income countries showed that most studies on AMS interventions report a positive effect for hospitalised patients.[17] However, only two low-income countries were represented in this review and no recommendations could be made for AMS interventions because of the variation of strategies reviewed.[17] Another review that studied the scope of AMS in children did not specifically identify interventions in low resource settings.[22] There is a paucity of data to guide AMS interventions in low-income countries although they face the most risk of infectious disease and AMR.[4]

To address this, our systematic review aims to identify protocols, policies or procedures for AMS in hospital settings in least developed countries, defined by the Development Assistance Committee (DAC) criteria.[23] The aim of this systematic review is to answer the following research question: What are the interventions (protocols, policies or procedures) for antimicrobial stewardship for hospitalised patients of all ages in least developed and low-income countries? Second, we aim to describe the impact of antimicrobial stewardship interventions on prescribing practices, clinical outcomes and AMR in least developed and low-income countries.

## METHODS
### Protocol and registration
The systematic review protocol was registered PROSPERO, with the International Prospective Register of Systematic Reviews. It will be conducted according to Preferred Reporting Items for Systematic Reviews and Meta-Analyses (PRISMA) checklist.[24] Amendments that may be made at any stage of the study will be reflected in amendments to the register.

### Study design
We will review AMS interventions in DAC least and low-income settings. We expect significant variation in the types of interventions studied and, in the methods, populations studied. Therefore, given anticipated limited data, there may not be sufficient studies to conduct a meta-analysis of interventions and outcomes.

### Eligibility criteria
#### Population
The population under study are neonates, children and adults in least developed and low-income countries as classified by DAC criteria.[23] We will include studies about hospital-based AMS approaches including surgical patients. Trials on long-term patients and community-based patients will be excluded (table 1).

#### Intervention
The intervention of interest is all Antimicrobial stewardship approaches, policies, practices or regulations in place to ensure judicious prescription of antimicrobials (table 1). We will include studies for which the intervention is clearly applied to the specified population. Reviews of interventions will not be included.

Interventions will be described as structural, enabling, persuasive, restrictive or combined according to the Cochrane Interventions to improve antibiotic prescribing practices for hospital inpatients.[25] Structural interventions include using digital records, rapid antimicrobial diagnostics or inflammatory markers to guide antimicrobial prescribing. Persuasive interventions include audits of prescription practices of healthcare workers and feedbacking information on where they can improve. Enabling interventions involve actively educating healthcare workers through treatment guidelines, training modules and reminders (posters or cards). Restrictive interventions involve authorising antimicrobial prescriptions, necessitating antibacterial sensitivity analyses or automatically discontinuing antimicrobial therapy for certain conditions after a specified period. Combined or bundled interventions will be those that combine at least two types of these interventions.

**Table 1** Systematic review inclusion and exclusion criteria

| PICO | Inclusion criteria | Exclusion criteria |
|---|---|---|
| Population | Studies on AMS, protocols or policies in DAC least developed and low-income countries<br>Hospitalised patients including surgical patients.<br>All age groups.<br>Studies on prescribers for hospitalised patients in DAC least developed countries | Trials on patients in the communities with no history of hospital admission during the period of the trial.<br>Patients classified by respective study to be in long-term care |
| Intervention | Antimicrobial stewardship approaches, programmes, policies where AMS is the sole intervention or at least one element of an intervention; and/or interventions targeting hospital hygiene or infection transmission control strategies. | Studies on viral, fungal or parasite infections where bacterial infection data cannot be extracted. |
| Comparison | None | None |
| Outcomes | Behavioural outcomes: changes in prescribing practices<br>Microbiological outcomes: bacterial infection confirmed by culture and sensitivity or by clinical suspicion. Confirmed microbiological infection with resistant bacteria at admission, during hospital stay or post discharge.<br>Clinical outcomes: treatment success (confirmed microbiological clearance or clinical improvement) length of stay in hospital or death. | Studies comparing the therapeutic effectiveness of one type of antimicrobial against another type. These are studies with the primary goal of determining superiority of one drug. |

AMS, antimicrobial stewardship; DAC, Development Assistance Committee; PICO, Participants Intervention Comparison Outcomes.

All interventions involving hygiene and sanitation will be described as infection prevention strategies. Studies with information on infection prevention will be included if they are incorporated as part of at least one type of AMS intervention described above.

### Comparator
There is no comparator for this systematic review.

### Outcome
The outcomes of interest will be categorised into behavioural, clinical and microbiological outcomes (table 1). Behavioural outcomes include changes in prescribing practices among medical personnel and compliance to AMS policies or protocols. Where available, a metric will be used to describe changes in prescribing practices, for example, daily doses per 100 bed-days.

Clinical outcomes include the following: treatment success will be defined as the successful eradication of a specified pathogen from an individual by microbiological confirmation or by clinical improvement described by a study physician. Clinical outcomes will also include mortality or length of stay in hospital or duration of antimicrobial use.

Microbiological outcomes include isolation of bacteria resistant to at least one antibiotic. It is expected that clinical suspicion of bacterial infection is a major indication for antibiotic prescription in the population of interest. Therefore, we will describe bacterial infection as clinical suspicion or confirmed microbiological infection. Confirmed microbiological infection will apply to studies that performed culture and or susceptibility testing. In this way, we intend to identify AMS protocols, which were applied for treatment of confirmed and suspected infections. Infection will apply to patients who had a bacterial infection as their primary illness or as a comorbidity of an underlying condition.

### Information sources
The data sources include PubMed, Ovid Medline, Ovid EMBASE, WHO Library Database (WHOLIS), African Index Medicus (AIM) and Cochrane Central Register of Controlled Trials (CENTRAL). We will study bibliographies in studies found to create a comprehensive list of relevant studies that might have been missed during the initial database search. Where necessary, we will contact authors for supporting information.

### Types of studies
We will include randomised controlled trials (RCT), before–after (CBA, controlled and non-controlled) interrupted time series (ITS, controlled and non-controlled) and cohort studies. We will also include qualitative studies that describe interventions targeting behavioural change and other relevant AMS strategies.

### Search strategy
Two independent reviewers will conduct literature search using the prespecified terms.

### Population
Low-income country or least developed country and hospital or primary care or community hospital or district hospital or tertiary hospital or tertiary care and adults or children or neonates or obstetrics or women.

### Intervention
Antimicrobial stewardship or antimicrobial stewardship protocols or rapid antimicrobial diagnostics or antimicrobial training.

## Outcome

Prescription practices or compliance or treatment success or mortality or length of hospital stay or AMR or first-line antibiotics or second-line antibiotics or multidrug-resistant bacterial infection.

During the review of titles and abstracts, reviewers will include publications on AMR and antimicrobial stewardship in the countries listed in the DAC criteria as least developed or low income. Studies will be excluded if they are about animal studies or are exclusively primarily about viral, fungal or parasite infections, with no implicationsof, or data on bacterial infections that can be extracted. We will limit this systematic review to studies published in English. We will only include articles from studies between the year 2000 and 2020. The list of articles collected by the two reviewers will be consolidated and duplicates will be removed.

During the second phase of the review, full-text review and analysis will be done. Studies included will need to have occurred within the population of interest and the interventions need to be clearly applicable to specified settings.

A third reviewer will resolve discrepancies in selection and their decision will be final. Full-text review will be conducted for the selected literature.

### Data extraction, synthesis and management

Data extraction will be done using the Cochrane collaboration data collection form for intervention reviews.[26] This form is a comprehensive assessment of full-text reviews. Qualitative data will be extracted through a form adapted from the Cochrane guidelines for extracting qualitative information.[27] The reasons for exclusion at this stage will be recorded in these forms. The process of data selection leading up to data synthesis will be presented using the PRISMA flowchart.[28] Due to the expected variation in methods and outcomes, a meta-analysis will not be performed.

Information will be categorised according to the demographic group, that is, neonates, children under 18, adults and obstetric patients. Due to the expected variation in methods and reporting practices, a narrative approach will be used to synthesise and report the data. Outcomes will be categorised according to the target of the intervention. Emerging themes from the data will be presented. A data summary table will be created to demonstrate major themes and interventions from included studies.

Before beginning the systematic review, two authors will pilot the data extraction form independently for five full review papers. Discrepancies in data collection will be consolidated and the data collection form will be adapted to reflect all information required.

### Assessment of risk of bias

This systematic review will be conducted according to the PRISMA guidelines.[29]

The assessment of risk of bias will be done according to the Integrated quality Criteria for Review of Multiple Study Designs (ICROMS).[30] We expect that multiple study designs could be used to describe and assess AMS strategies in low resource settings. The ICROMS tool allows for a comprehensive assessment of the quality criteria for Randomized Controlled Trials, Interupted Time Series, Controlled Before-After, Non-Controlled Before-After cohort and qualitative studies.[30] This allows for the inclusion of qualitative studies, which may assess behavioural change in prescribing practices as is important to AMS.

This tool assesses the quality of studies on the social and contextual determinants of an intervention and outcome. ICROMS was adopted for systematic reviews and meta-analysis in infection prevention and antimicrobial prescribing.[30]

Using the ICROMS tool, each study will be assessed for universally acceptable standards for its respective study type. This means randomised controlled studies will be assessed according to their own study type and standards and qualitative studies will also have their own criterion. Seven dimensions describe the quality criteria each study type will be assessed for. These are clear aims and justification, managing bias in sampling or between groups, managing bias in outcome measurements and blinding, managing bias in follow-up, managing bias in other study aspects, analytical rigour and managing bias in reporting/ethical considerations. Each type of study will then be assessed on its quality criteria among 33 criteria.[30]

Using the ICROMS tool, methodological robustness of a study is analysed by assessing whether the study meets the prespecified mandatory criterion and its type of study. Studies that do not meet the mandatory criteria will be excluded.

The scientific validity of each study will be assessed by calculating a minimum score of criteria met within the quality dimensions.[30] Each criterion met receives 2 points, if it is unclear whether the criteria is met, it is given 1 point and no points are accorded for study criteria that are clearly not met. The total quality score for each study is compared with the minimum score of its study type. The minimum score for each type of study represents 60% of the criteria required to demonstrate that the study was scientifically valid and that the outcomes are reliable.[30] The minimum scores range from 22 points for RCT, cluster RCT, Non-Controlled Interrupted Time Series and Non-Controlled Before-After, 18 points for Controlled Before-After, Controlled Interrupted Time Series and cohort studies and 16 points for qualitative studies.[30] Studies not meeting the minimum score will be excluded.

The adapted ICROMS template is shown in online supplemental appendix. This template includes a summary table for the risk of bias analysis of all studies. We will reference the detailed ICROMS protocol[30] when deciding on the score for each criterion.

## Ethics and dissemination

We do not expect to use patient data for this review and will not require ethical clearance. We will use the results from this review to design candidate interventions for AMS practices that can be tested this low resource setting. Our AMS protocol design will involve meeting with relevant Ministry of Health and Chikwawa Clinical officials to discuss which elements of our findings can be incorporated into an AMS strategy. This review will specifically inform the development of AMS practices and policies to be tested by the Paediatric Research in Antimicrobial Stewardship and Management network in Chikwawa district in Malawi. Malawi is a southern-African country classified as a DAC least developed country, and the incidence of AMR is rising.[23 31] Finding ways to promote AMS in this setting can be useful for similar settings as well. We will publish findings from this review in peer-reviewed journals and present our research to local and international conferences.

**Contributors** P-YIT is the guarantor of the review. GWM and P-YIT conceived the topic and the structure of the protocol. GWM wrote the first draft of the protocol and will coordinate subsequent systematic review activities. P-YIT edited the manuscript and MM read and provided feedback.

**Funding** The sponsor for this project is the Liverpool School of Tropical Medicine and it was funded by the Wellcome Trust. This work was supported by a Wellcome Trust Programme Grant (Grant number 091909/Z/10/Z) and the MLW Programme Core Award (Grant 206545/Z/17/Z) from the Wellcome Trust.

**Disclaimer** The Funders had no role in conceiving the project and in writing the manuscript. The funders will not provide input to collecting and analysing the data or in disseminating the results.

**Competing interests** None declared.

**Patient consent for publication** Not required.

**Provenance and peer review** Not commissioned; externally peer reviewed.

**ORCID iDs**
Grace Wezi Mzumara http://orcid.org/0000-0001-8132-1620
Pui-Ying Iroh Tam http://orcid.org/0000-0002-3682-8892

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
