## [Reviewer comments · BMJ Open]

ARTICLE DETAILS

TITLE (PROVISIONAL)	Antimicrobial Stewardship Interventions in Least Developed and Low-Income Countries: A Systematic Review Protocol
AUTHORS	Mzumara, Grace; Mambiya, Michael; Iroh Tam, Pui-Ying

VERSION 1 – REVIEW

REVIEWER	Herbison, Peter University of Otago, Preventive and Social Medicine
REVIEW RETURNED	31-Jan-2021

GENERAL COMMENTS	This is a protocol for an unusual systematic review. But it is lacking detail about a search strategy. It will be a difficult search for this project, so what they have given may be deemed a first pass. I am a bit concerned about the system for judging study quality (ICROMS). It is based on a good idea, to have a system that is applicable to many study types. But it relies on a scoring system, and these have been shown not to do well. For example, the score relies on every element in the scoring system being weighted equally, which is usually wrong because some elements are more important than others. But it seems that ICROMS is just used to see if the study is reliable enough to be included so used with some caution it may be all right. Because of my interest in study quality I looked up the paper. I found that the reference to it (30) is wrong. It is missing one of the authors, and the year is given as 2015 but should be 2016. Many other references have things omitted, such as journal names. The references should be checked carefully and put into BMJ Open standard form. The authors use different terms for the same thing. They use "resistance to antimicrobials", "antibacterial resistance" and "antibiotic resistance". Using the same wording throughout should decrease confusion. I am not sure why reviews of interventions will not be included. I would think that at least the reference list in these would be of interest. There are some acronyms that are not clearly defined. For example before-after studies are denoted as "CBA" which I think is "Controlled before-after". These should be checked.
--

REVIEWER	Padigos , Junel Sunshine Coast Hospital and Health Service, Intensive Care Unit
REVIEW RETURNED	15-Feb-2021

GENERAL COMMENTS	The reviewer provided a marked copy with additional comments. Please contact the publisher for full details.
--

REVIEWER	Kpokiri, Eneyi London School of Hygiene & Tropical Medicine, Faculty of Infectious and Tropical Diseases
REVIEW RETURNED	15-Feb-2021

GENERAL COMMENTS	i commend the authors for undertaking a systematic review in this important topic. this protocol have been written and presented well. I will
--

VERSION 1 – AUTHOR RESPONSE

Reviewer: 1

Prof. Peter Herbison, University of Otago

I am a bit concerned about the system for judging study quality (ICROMS). It is based on a good idea, to have a system that is applicable to many study types. But it relies on a scoring system, and these have been shown not to do well. For example, the score relies on every element in the scoring system being weighted equally, which is usually wrong because some elements are more important than others. But it seems that ICROMS is just used to see if the study is reliable enough to be included so used with some caution it may be all right.

Response: Thank you for making this observation. We will consider your insights during study quality assessment in the paper.

Because of my interest in study quality I looked up the paper. I found that the reference to it (30) is wrong. It is missing one of the authors, and the year is given as 2015 but should be 2016. Many other references have things omitted, such as journal names. The references should be checked carefully and put into BMJ Open standard form.

Response: Thank you for your response and for noting the errors in referencing

Action: We have reviewed all references and have made necessary corrections to all references. Below is the reference we used for this paper: 'Zingg W, Castro-Sanchez E, Secci F V, et al. Innovative tools for quality assessment : integrated quality criteria for review of multiple study designs. Public Health. 2015;133:19–37.doi: 10.1016/j.puhe.2015.10.012'
<https://pubmed.ncbi.nlm.nih.gov/26704633/>

The authors use different terms for the same thing. They use "resistance to antimicrobials", "antibacterial resistance" and "antibiotic resistance". Using the same wording throughout should decrease confusion.

Response: Thank you for your asking for this clarification

Action: We have decided to use antimicrobial resistance throughout the paper

I am not sure why reviews of interventions will not be included. I would think that at least the reference list in these would be of interest.

Response: Thank you for your response.

Action: We included reviews of interventions in this systematic review

There are some acronyms that are not clearly defined. For example before-after studies are denoted as "CBA" which I think is "Controlled before-after". These should be checked.

Response: Thank you for your review and comments.

Action: We have clarified controlled before-after as CBA

Reviewer: 2

Dr. Junel Padigos , Sunshine Coast Hospital and Health Service

Response: Thank you for your response. We have reviewed the paper to change wordings and errors.

Reviewer: 3

Dr. Eneyi Kpokiri, London School of Hygiene & Tropical Medicine

Response: Thank you for your response, we appreciate your feedback